# Regulation of plant immune receptor accumulation through translational repression by a glycine-tyrosine-phenylalanine (GYF) domain protein

Zhongshou Wu[1,2†], Shuai Huang[1,2†‡], Xiaobo Zhang[1,3†], Di Wu[2], Shitou Xia[3*], Xin Li[1,2*]

[1]Michael Smith Laboratories, University of British Columbia, Vancouver, Canada; [2]Department of Botany, University of British Columbia, Vancouver, Canada; [3]Hunan Provincial Key Laboratory of Phytohormones, Hunan Agricultural University, Changsha, China

**Abstract** Plant immunity is tightly regulated to ensure proper defense against surrounding microbial pathogens without triggering autoimmunity, which negatively impacts plant growth and development. Immune receptor levels are intricately controlled by RNA processing and post-translational modification events, such as ubiquitination. It remains unknown whether, and if yes, how, plant immune receptor homeostasis is regulated at the translational level. From a *mutant, snc1-enhancing* (*muse*) forward genetic screen, we identified MUSE11/EXA1, which negatively regulates nucleotide-binding leucine-rich repeat (NLR) receptor mediated defence. EXA1 contains an evolutionarily conserved glycine-tyrosine-phenylalanine (GYF) domain that binds proline-rich sequences. Genetic and biochemical analysis revealed that loss of *EXA1* leads to heightened NLR accumulation and enhanced resistance against virulent pathogens. EXA1 also associates with eIF4E initiation factors and the ribosome complex, likely contributing to the proper translation of target proteins. In summary, our study reveals a previously unknown mechanism of regulating NLR homeostasis through translational repression by a GYF protein.

*For correspondence:
xstone0505@yahoo.com (SX);
xinli@interchange.ubc.ca (XL)

[†]These authors contributed equally to this work

**Present address:** [‡]Howard Hughes Medical Institute, Yale University School of Medicine, New Haven, United States

**Competing interests:** The authors declare that no competing interests exist.

## Introduction

Plants have evolved complex defense systems to recognize and respond to microbial pathogens. Plant plasma membrane localized pattern recognition receptors (PRRs) can perceive conserved pathogen-associated molecular patterns (PAMPs), e.g. bacterial flagellin, fungal chitin and damage-associated molecular patterns (DAMPs) resulting from damaged tissues (*Dangl et al., 2013*; *Ferrari et al., 2013*). The defence responses activated by PRRs is defined as PAMP-triggered immunity (PTI), which usually leads to the production of reactive oxygen species, the deposition of callose and the increased expression of defense genes (*Macho and Zipfel, 2014*). However, successful pathogens are able to deliver effectors to suppress PTI and promote infection. To overcome susceptibility, plants have evolved resistance proteins (R proteins), which can activate strong immune responses upon recognition of cognate microbial effectors (*Li et al., 2015*). The majority of *R* genes encode proteins with nucleotide-binding leucine-rich repeat domains (NLR), which can also be found in animal immune receptors such as Nod proteins. Typical plant NLRs can be further classified into Toll-like/Interleukin 1 receptor (TIR)-type NLRs (TNLs) and coiled-coil NLRs (CNLs) based on their corresponding N-terminal domains. Plant NLR-mediated immune responses often culminate in a hypersensitive response (HR), which is a programmed cell death event believed to restrict pathogen

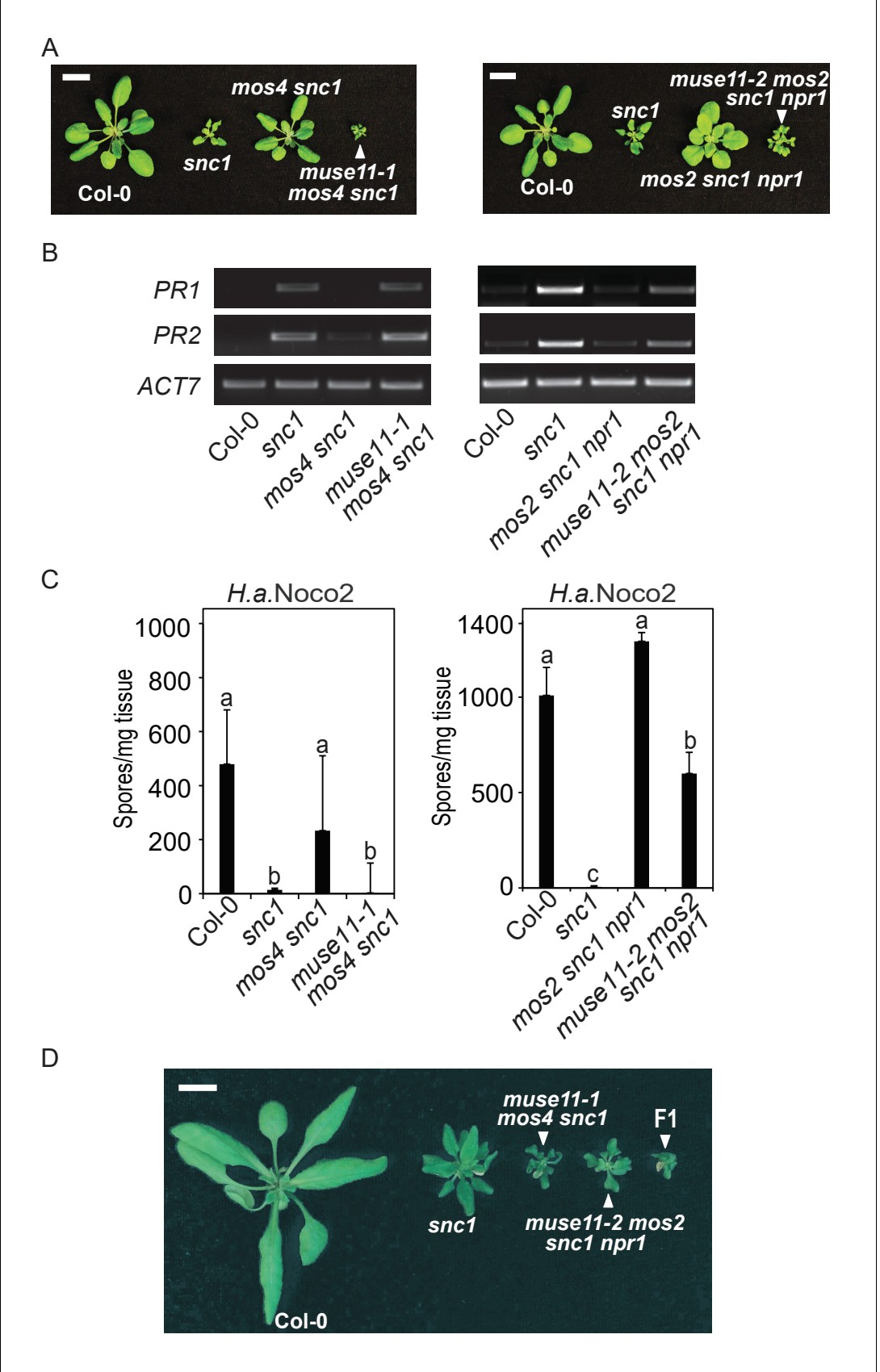

**Figure 1.** Characterization of *muse11-1 mos4 snc1* and *muse11-2 mos2 snc1 npr1* mutants. (**A**) Morphology of four-week-old plants of the indicated genotypes. Plants were grown at 22°C under long day conditions. Bar = 1 cm. (**B**) *PR1* and *PR2* gene expression in the indicated plants as determined by RT-PCR. (**C**) Quantification of *H.a.* Noco2 sporulation on the indicated plants seven days post inoculation (dpi) with $10^5$ spores per ml water. One-way ANOVA was used to calculate the statistical significance among genotypes. Different letters indicate statistical differences (p<0.05). Three independent experiments were carried out with similar results. Bars represent means ± SD (n = 4). (**D**) Morphology of four-week-old Col-0, *snc1*, *muse11-1 mos4 snc1* and *muse11-2 mos2 snc1 npr1* plants with an F1 progeny from the cross between *muse11-1 mos4 snc1* and *muse11-2 mos2 snc1 npr1*. Bar = 1 cm.

growth at the site of infection (*Dangl et al., 2013*). Although many *R* genes have been cloned from plants, the molecular mechanisms of most NLR-mediated signal transduction are unclear.

NLR-mediated defense has to be tightly controlled as constitutively activated NLRs can lead to autoimmune phenotypes, whereas insufficient or non-functional NLRs can cause susceptibility to specific pathogens. NLR homeostasis control is critical in producing the proper levels of defence output. For example, the stability of TNL protein Suppressor of *npr1-1*, Constitutive 1 (SNC1) is enhanced in the autoimmune *snc1* mutant, leading to constitutive defense activation and dwarfism (*Zhang et al., 2003*; *Cheng et al., 2011*; *Gou et al., 2012*). The regulation of plant NLR homeostasis occurs at multiple levels (*Li et al., 2015*). At the transcriptional level, the epigenetic regulation, e.g. DNA methylation and histone modification events, can control the transcript levels of *NLR* genes. The alteration of DNA methylation in the *SNC1* genomic region reduces the expression of *SNC1* and *RPP4*, located at the same *R* gene cluster as *SNC1* (*Xia et al., 2013*; *Zou et al., 2014*). At the post-transcriptional levels, the MOS4-associated complex (MAC), consisting MOS4, AtCDC5, PRL1, MAC3 and MAC5 and other proteins, e.g. MOS12, MOS2 and NRPC7, affect the splicing pattern of *NLR* genes. Different versions of NLR proteins encoded by the resulting spliced mRNAs likely affect the levels of the functional NLR proteins (*Zhang et al., 2005*; *Palma et al., 2007*; *Monaghan et al., 2009*, *2010*; *Xu et al., 2011*, *2012*; *Copeland et al., 2013*; *Johnson et al., 2016*). At the post-translational level, a number of regulators involved in NLR folding and turnover have been identified. Molecular chaperones, including HSP90, SGT1 and RAR1, are required for the proper accumulation of some NLRs (*Shirasu, 2009*; *Huang et al., 2014a*). The ubiquitin-mediated proteasome system controls the turnover of NLRs. For example, Arabidopsis SCF[CPR1] E3 ligase is responsible for regulating the turnover of SNC1 and RPS2 (*Cheng et al., 2011*; *Gou et al., 2012*). Barley RING-type E3 ligase MIR1 regulates CNL MLAs levels via the ubiquitin proteasome system (*Wang et al., 2016*). Besides the SCF[CPR1] complex, the E4 factor, a TRAFasome with MUSE13/14, and CDC48A can also modulate the NLR levels (*Huang et al., 2014b*; *Copeland et al., 2016*; *Huang et al., 2016*).

An alternative mechanism to control protein homeostasis is through translational processes. In this study, we report the isolation, identification, characterization and functional analysis of *MUSE11* (Mutant, *snc1*-enhancing 11) from the MUSE forward genetic screens. The *muse11* mutants were found to be novel alleles of *exa1*. *EXA1* encodes a GYF domain containing protein. Biochemical analysis of the *exa1* mutants and the EXA1 interactors revealed that EXA1 likely regulates NLR levels through translational repression.

## Results

### Isolation, characterization and positional cloning of *muse11*

To identify novel negative regulators of plant immunity, we previously performed *muse* (mutant, *snc1*-enhancing) forward genetic screens to isolate *snc1* enhancers (*Huang et al., 2013*). Two allelic mutants, *muse11-1* and *muse11-2*, were isolated from the *mos4 snc1* and the *mos2 snc1 npr1* background, respectively. As with all other *muse* mutants, both *muse11* alleles enhance *snc1*-mediated autoimmunity in the *mos snc1* backgrounds, including stunted growth (*Figure 1A*), elevated defence marker *Pathogenesis Related* (*PR*) gene expression (*Figure 1B*) and enhanced disease resistance against the virulent oomycete pathogen *Hyaloperonospora arabidopsidis* (*H.a.*) Noco2 (*Figure 1C*). When the isolated mutants were backcrossed with *mos snc1* parents, the F1 plants were *mos snc1*-like, indicating that *muse11-1* and *muse11-2* are recessive. When *muse11-1 mos4 snc1* was crossed

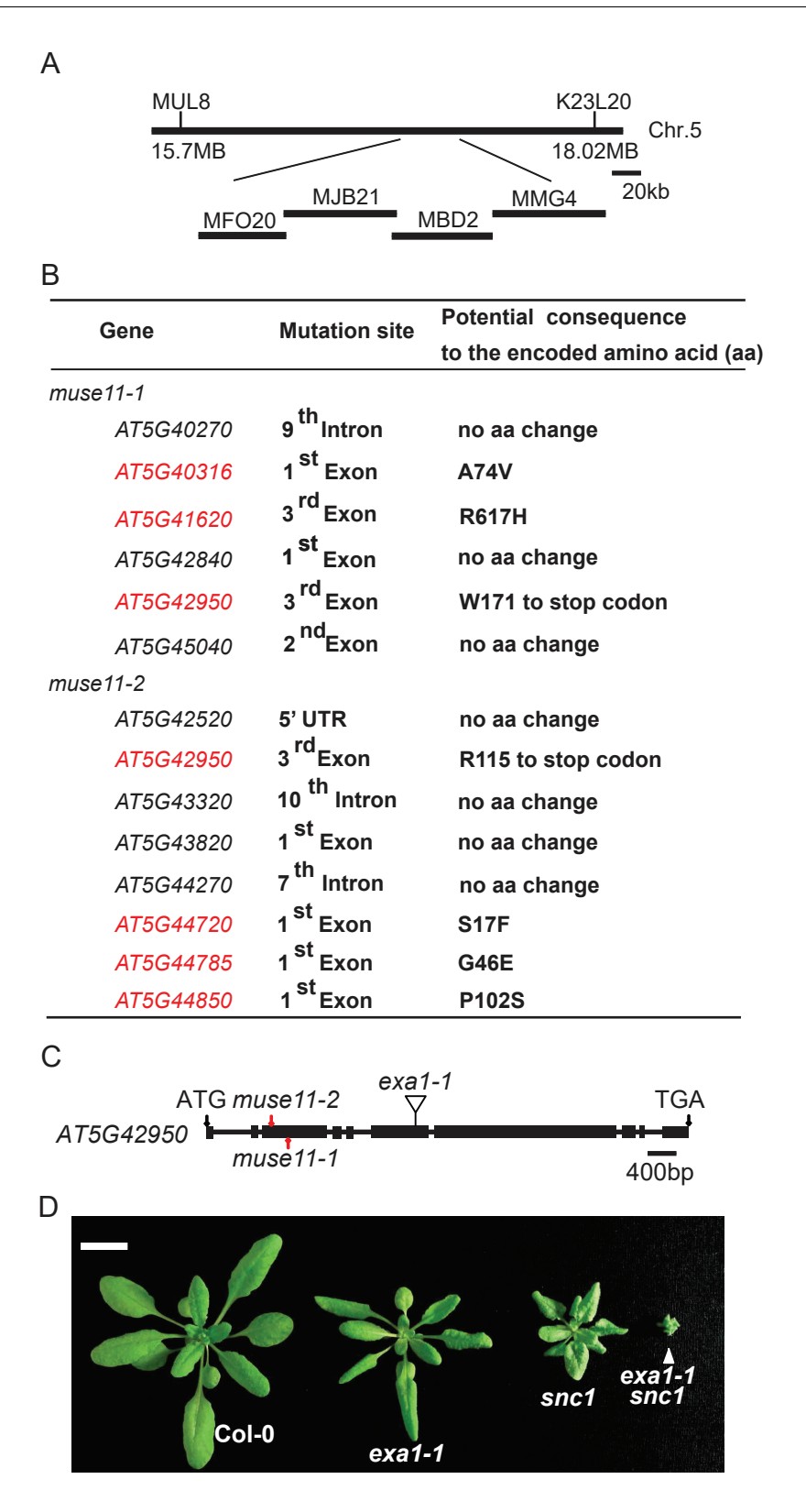

**Figure 2.** Map-based cloning of *muse11*. (**A**) Map position of *muse11-1* and *muse11-2* on chromosome 5. BAC clones are indicated. (**B**) Mutations identified in the mapped *muse11-1* and *muse11-2* region through next-generation resequencing. Genes carrying non-synonymous mutations are indicated in red fonts. (**C**) Gene structure

*Figure 2 continued on next page*

*Figure 2 continued*

of *MUSE11* (*AT5G42950*). Boxes indicate exons and lines represent introns. The two black arrows indicate the start and stop codons, respectively. The two mutations found in *muse11-1* and *muse11-2* are labeled with red arrows. The insertion site in *exa1-1* (SALK_005994) is illustrated in exon 6. (**D**) Morphology of four-week-old Col-0, *exa1-1*, *snc1* and *exa1-1 snc1* plants. Bar = 1 cm.

with *muse11-2 mos2 snc1 npr1*, all F1 progeny showed dwarfism similar to both parent lines, indicating failed complementation (*Figure 1D*). Therefore *muse11-1* and *muse11-2* likely carry mutations in the same gene.

Linkage analysis located both *muse11-1* and *muse11-2* to the same region of bottom arm of Chromosome 5 (*Figure 2A*). Comparison of the genome re-sequenced *muse11-1 mos4 snc1* and *muse11-2 mos2 snc1 npr1* sequences in the *muse11* region revealed different mutations in only one common gene, *AT5G42950* (*Figure 2B*), which were not present in either the *mos4 snc1* or *mos2 snc1 npr1* parent line. Sequence analysis revealed that the two *muse11* alleles harbour different early stop codon mutations, presumably causing truncations of the encoded protein (*Figure 2B*). Therefore *MUSE11* is most likely *AT5G42950*. This gene was recently named as *EXA1* (*Essential for poteX-virus Accumulation*), as loss of EXA1 causes enhanced resistance to viruses (*Hashimoto et al., 2016*).

To further confirm that *MUSE11* is *AT5G42950*, we crossed a T-DNA allele of *AT5G42950*, *exa1-1* (SALK_005994), with *snc1*. SALK_005994 contains a T-DNA insertion in the sixth exon (*Figure 2C*), presumably knocking out *AT5G42950*. Homozygous T-DNA lines were identified by PCR and crossed with *snc1*. As shown in *Figure 2D*, *exa1-1* greatly enhanced the *snc1* phenotypes as the *muse11* alleles. Thus, we concluded that *MUSE11* is *AT5G42950*, and therefore renamed *muse11-1* and *muse11-2* as *exa1-2* and *exa1-3*, respectively. In this study, *exa1-1* and *exa1-2* were further characterized in detail.

## *exa1* single mutant analysis

Both *exa1-1* and *exa1-2* single mutant plants are smaller in size compared with WT and display a curly leaf phenotype similar to *snc1*, suggestive of autoimmunity (*Figure 3A*). RT-PCR analysis showed that the expression of *PR1* and *PR2* is significantly increased in both *exa1-1* and *exa1-2* plants (*Figure 3B*). When challenged with virulent pathogens, both alleles cause enhanced resistance against *H.a.* Noco2 and bacteria *Pseudomonas syringae* pv. *maculicola* (*P.s.m.*) ES4326 (*Figure 3C and D*). Collectively, these phenotypic analyses of the knockout mutants indicate that EXA1 plays a negative role in immunity; loss of EXA1 leads to enhanced resistance.

## *EXA1* encodes a GYF domain protein and is conserved in plants

Protein sequence analysis showed that *EXA1* encodes a large protein of 1714 amino acids, with a predicted eukaryotic translation initiation factor 4E (eIF4E) binding motif and a glycine-tyrosine-phenylalanine (GYF) domain (*Figure 4A*). GYF domains are molecular adaptors that bind to proline-rich sequences (*Kofler and Freund, 2006*). The GYF domain in EXA1 is composed of 57 amino acid residues from position 546–603, close to its N-terminus. Phylogenetic analysis showed that GYF domains are found in all eukaryotes (*Kofler and Freund, 2006*) (*Figure 4B*). However, when EXA1 full-length protein was analyzed, its putative orthologs are restricted to the plant lineage (*Figure 4C*). In *Arabidopsis*, there are eight GYF domain-containing proteins (*Figure 5A and B*). The SWIB/PHD/GYF clade proteins, exemplified by NERD (Needed for RDR2-independent DNA methylation; AT2G16485), seem to be involved in transcription and chromatin modifications, participating in chromatin-based RNA silencing (*Pontier et al., 2012*). However, the roles of the EXA1 clade proteins in plants remain elusive.

*AT1G24300* and *AT1G27430*, the two closest paralogs of *EXA1*, share 45% and 43% sequence similarity with *EXA1*, respectively (*Figure 5A*). Phylogenetic analysis revealed that these two genes likely have evolved independently from *EXA1* (*Figure 4C*). To examine whether these two EXA1 paralogs also exert negative roles in *snc1*-mediated immunity, we carried out double mutant analysis by crossing the T-DNA alleles of these two genes with *snc1*. As shown in *Figure 5C and D*, homozygous exonic T-DNA alleles of *AT1G24300* (SALK_ 058114) and *AT1G27430* (SALK_035304) are indistinguishable from WT and their double mutant plants with *snc1* display *snc1*-like morphology. Taken

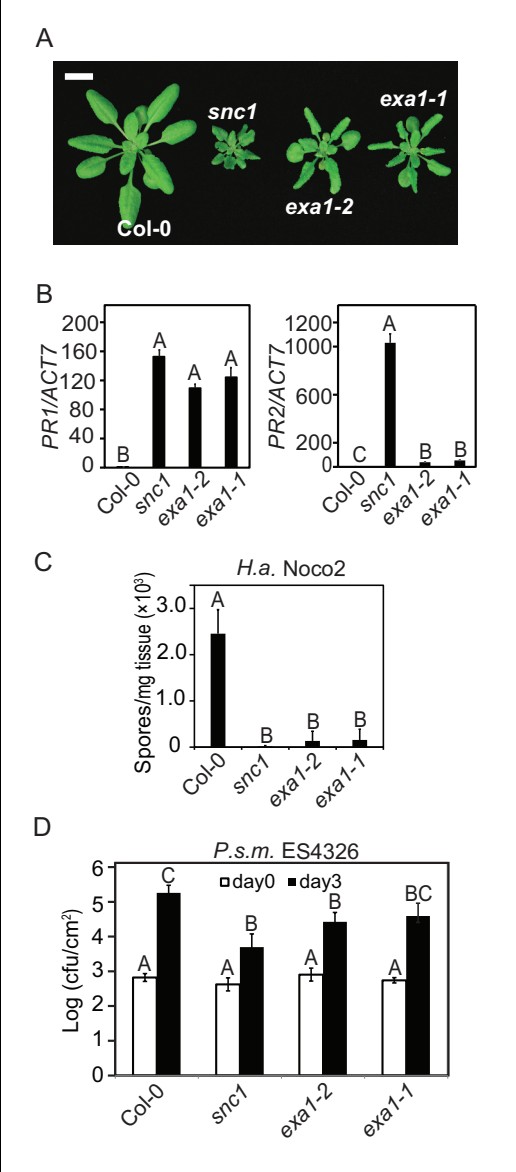

**Figure 3.** Characterization of the *exa1* single mutants. (A) Morphology of four-week-old Col-0, *snc1*, *exa1-2* and *exa1-1* plants. Bar = 1 cm. (B) Relative expression of *PR-1* and *PR-2* in four-week-old soil-grown Col-0, *exa1-2* and *exa1-1* plants as determined by RT-PCR. *Actin7* was used to normalize the transcript levels. One-way ANOVA was used to calculate the statistical significance among genotypes, which is indicated by different letters (p<0.01). Two independent experiments were carried out with similar results. Bars represent means ± SD (n = 3). (C) Quantification of *H.a.* Noco2 sporulation on the indicated genotypes. One-way ANOVA was used to calculate the statistical significance among genotypes, which is indicated by different letters (p<0.01). Three independent experiments were carried out with similar results. Bars represent means ± SD (n = 4). (D) Growth of *P.s.m.* ES4326 on four-week-old leaves of the indicated

*Figure 3 continued on next page*

together, *EXA1* seems to be a single-copy gene involved in the negative regulation of plant immunity.

## EXA1 localizes in the cytosol

In yeast and animals, GYF domain proteins have been implicated to play diverse roles such as mRNA splicing and translational control depending on where they reside in the cell (*Ash et al., 2010*). For example, the GYF protein CD2BP2 (CD2-binding protein 2) localizes in the nucleus and participates in mRNA splicing (*Kofler et al., 2004*). However, another GYF protein SMY2 (suppressor of myosin 2) in yeast seems to play diverse roles including in ribosome biogenesis and translational control in the cytosol (*Ash et al., 2010*; *Okano et al., 2015*).

To analyze the functions of EXA1, we first examined its subcellular localization. EXA1 is predicted to localize to the chloroplast on The Arabidopsis Information Resource (TAIR). However, we did not identify any signal peptide using diverse web-based Signal Peptide Prediction software (data not shown), suggesting that the annotation on TAIR could be inaccurate. We also did not find any transmembrane domains or nuclear localization signals. We thus transformed a wild-type copy of EXA1 translationally fused with a GFP tag at the C-terminus driven by the *EXA1* native promoter into the *exa1-2* single mutant. As shown in *Figure 6*, EXA1-GFP largely complemented the autoimmune phenotypes of *exa1-2*, including stunted growth (*Figure 6A*) and enhanced resistance to *H.a.* Noco2 and *P.s.m.* ES4326 (6B and 6C), suggesting that *EXA1-GFP* is functional and likely localizes to its native sites. Western blot analysis revealed that the full-length EXA1-GFP protein is successfully expressed in the transgenic lines (*Figure 6D*). Confocal microscopy examination revealed that EXA1-GFP localizes to the cytosol, but is absent from the nucleus of *Arabidopsis* root cells (*Figure 6E*), indicating that EXA1 is unlikely to be involved in mRNA splicing as with CD2BP2. Rather, it may play other roles, such as translational control, as with SMY2. In agreement with this observation, the GYF domain in EXA1 is more similar to SMY2 than with CD2BP2 (*Figure 4B*).

## EXA1 negatively regulates SNC1 protein accumulation

We hypothesized that EXA1 may exhibit its negative role in SNC1-mediated immunity at the translational level based on four reasons: (1) EXA1 possesses an N-terminal eIF4E binding motif,

*Figure 3 continued*

genotypes at 0 and 3 dpi with bacterial inoculum of OD$_{600}$ = 0.001. One-way ANOVA was used to calculate the statistical significance among genotypes, which is indicated by different letters (p<0.01). Three independent experiments were carried out with similar results. Bars represent means ± SD (n = 4).

where eIF4E is involved in translation initiation (*Hashimoto et al., 2016*); (2) translational control plays a critical role in maintaining protein homeostasis (*Lykke-Andersen and Bennett, 2014*); (3) EXA1 resides in the cytosol, where translation occurs; and (4) the *snc1* enhancer screen enables isolations of mutants defective in SNC1 homeostasis control (*Huang et al., 2013*; *Li et al., 2015*). To test this hypothesis, we first examined whether *exa1* affects SNC1 protein levels. As shown in *Figure 7A and B*, we consistently observed significantly increased SNC1 levels in both *exa1-1* and *exa1-2* single mutant plants compared with WT. Similarly, immunoblot analysis detected much higher SNC1 protein levels in the *exa1-1 snc1* double mutant when compared with *snc1* (*Figure 7A and B*), although the expression of *SNC1* is not affected (*Figure 7C*). These data suggest that EXA1 regulates SNC1 protein accumulation, likely through translational events, as it is predicted to associate with the translation machinery and localizes to the cytosol.

We further examined the effects of *EXA1* overexpression on SNC1 protein levels. When SNC1-HA was co-infiltrated with EXA1-FLAG in *Nicotiana benthamiana*, the overexpression of EXA1 renders lower SNC1 levels (*Figure 7D*). Importantly, *SNC1* transcript levels were not affected by EXA1 overexpression (*Figure 7E*). Consistent with the observations in tobacco, overexpression of the functional *EXA1-FLAG* in *snc1* partially suppresses the dwarf stature of *snc1* (*Figure 7F*) without affecting *SNC1* transcript levels (*Figure 7G*). However, EXA1-FLAG overexpression reduces the accumulation of SNC1 (*Figure 7H and I*), confirming the negative effects of EXA1 to SNC1 homeostasis.

## EXA1 negatively impacts the accumulation of other tested NLRs

Since the *exa1* single mutants accumulate more SNC1 protein and exhibit enhanced disease resistance phenotype, we asked whether the autoimmune phenotype of *exa1* is specific to SNC1. When double mutant *exa1-1 snc1-r1* (*Zhang et al., 2003*) was created, the plants are of intermediate size between *exa1-1 and snc1-r1* (*Figure 8A and B*). The autoimmunity of *exa1-1* against *H.a.* Noco2 was also partially suppressed by *snc1-r1* (*Figure 8C*), indicating that the enhanced immunity of *exa1* is only partially dependent on SNC1.

As SNC1 is a typical TNL and most TNLs depend on EDS1 (*Aarts et al., 1998*), we further tested whether EXA1 relies on EDS1 by analysing *exa1-1 eds1* double mutant. As shown in *Figure 8A–C*, the size, fresh weight and enhanced resistance of *exa1-1* against *H.a.* Noco2 were only partially suppressed by the loss of EDS1, indicating that EXA1 may also affect additional EDS1-independent NLRs. Taken together, these data suggest that the over-accumulation of SNC1 in *exa1* is only partially responsible for its enhanced resistance phenotype.

To test the role of EXA1 in controlling other NLRs, we investigated the protein levels of three additional NLRs, RPS4-HA, RPM1-Myc, and RPS2-HA in the *exa1-1* single mutant (*Grant et al., 1995*; *Axtell and Staskawicz, 2003*; *Wirthmueller et al., 2007*). Interestingly, all three NLRs accumulated to significantly higher levels in *exa1-1* (*Figure 9A–C*), although their transcript levels are not affected (*Figure 9D–F* and *Figure 9—figure supplement 1*), indicating that EXA1 has a general role in negatively controlling NLR protein accumulation.

To further examine the biological role of EXA1 in R protein mediated immunity, we challenged *exa1* single mutant plants with *Pst* DC3000 expressing avirulent effectors, including AvrRps4, AvrRpt2, and AvrRpm1, which are recognized by TNL RPS4 and CNL RPS2 and RPM1, respectively. As shown in *Figure 9G–H*, reduced bacterial growth of both *Pst* DC3000 AvrRps4 and *Pst* DC3000 AvrRpm1 was observed in the *exa1* single mutants. However, even though heightened RPS2 protein level was detected in the *exa1* single mutant plants (*Figure 9C*), no enhanced resistance against *Pst* DC3000 AvrRpt2 was observed (*Figure 9I*). It is possible that the threshold for achieving resistance against AvrRpt2 is higher than the accumulated RPS2 in the *exa1* single mutant. Taken together, besides SNC1, EXA1 also contributes to the accumulation of other NLRs including RPS4, RPS2 and RPM1.

To further examine the specificity of EXA1 in protein homeostasis control, we tested the protein level of several PTI immune receptors and non-immune proteins in *exa1* mutants. FLAGELLIN-

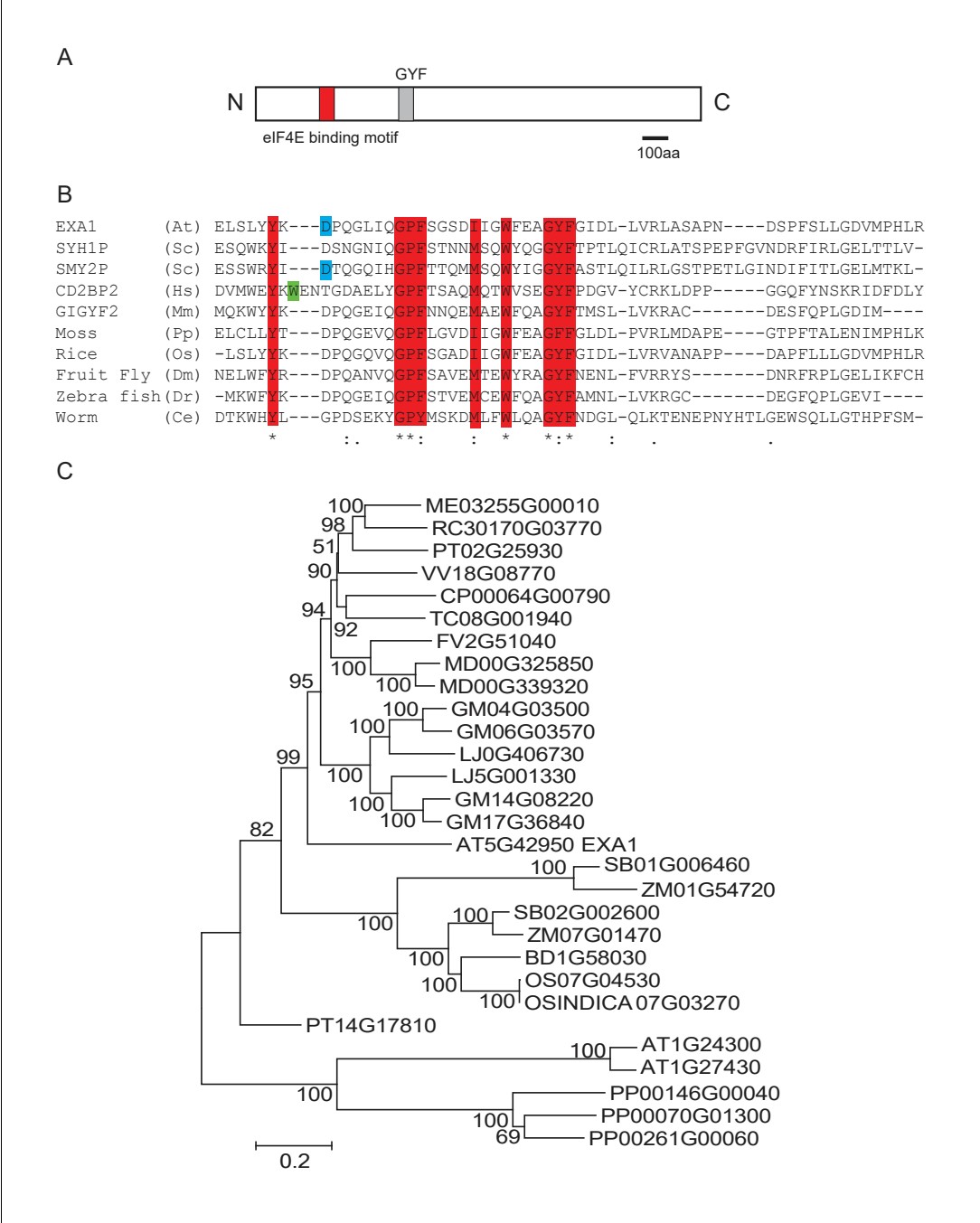

**Figure 4.** Phylogenetic analysis of EXA1 and its potential orthologs. (**A**) The predicted protein structure of EXA1. The GYF domain is indicated by a gray box. (**B**) Amino acid alignments of GYF domains from eukaryotes. At, Sc, Hs, Mm, Pp, Os, Dm, Dr and Ce stand for the following species: *A. thaliana*, *Saccharomyces cerevisiae* (SYH1P: AJW09613.1; SMY2P: AJQ12401.1), *Homo sapiens* (CD2BP2: ALQ33977.1), *Mus musculus* (GIGYF2: EDL40166.1), *Physcomitrella patens* (XP_001772117.1), *Oryza sativa* (XP_015646509.1), *Drosophila melanogaster* (NP_651950.3), *Danio rerio* (XP_009301398.1) and *Caenorhabditis elegans* (NP_001041150.1), respectively. Protein sequences were aligned using CLUSTAL. Residues that are characteristic of GYF domains are depicted as red letters. The green or blue letters represent residues conserved in either the CD2BP2 or SMY2 subfamily, respectively. (**C**) EXA1 and its potential orthologs in plant species were used to generate the phylogenetic tree. Putative EXA1 orthologs were obtained from Plaza using full-length protein sequences as input. Sequences were aligned using Muscle and a Neighbor Joining tree was constructed using MEGA 5.0 with the JTT model and 5000 bootstrap value.

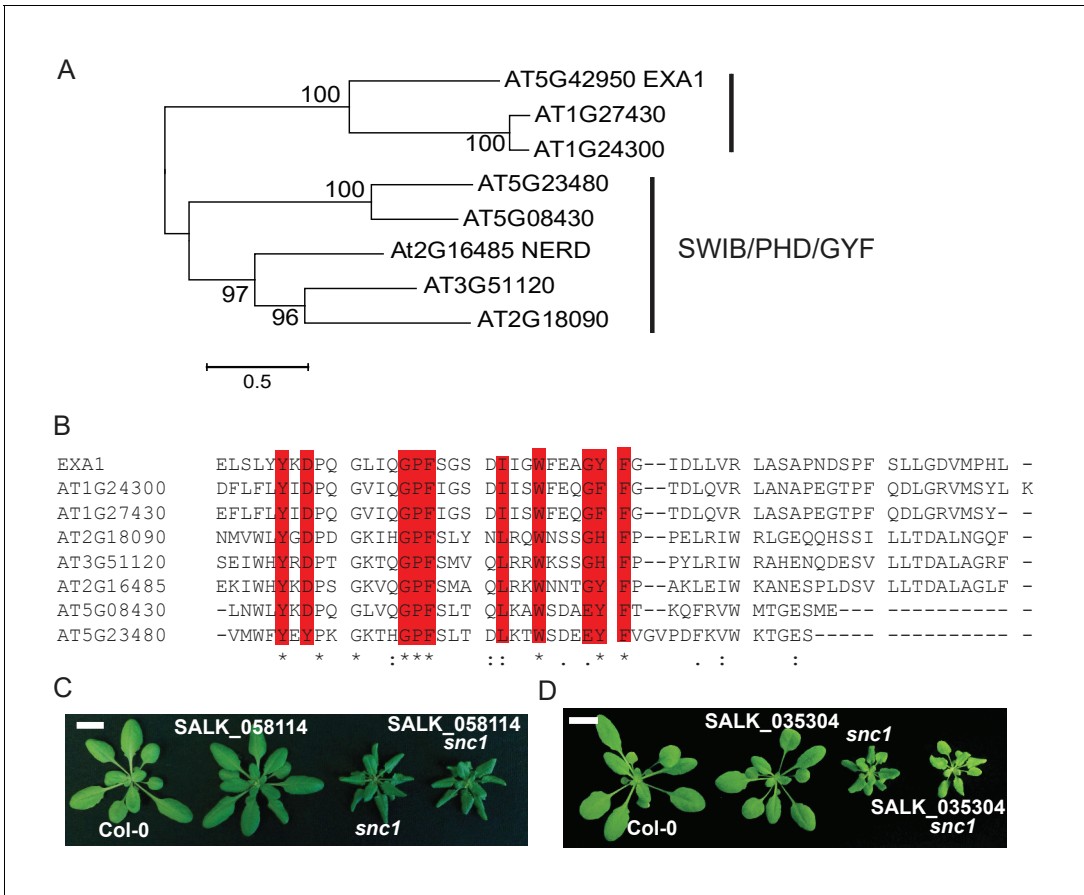

**Figure 5.** Sequence analysis of EXA1 and other GYF domain proteins in Arabidopsis. (**A**) A phylogenetic tree of all eight GYF-containing proteins in Arabidopsis. (**B**) Protein alignment of all GYF domains of *Arabidopsis* GYF proteins. Residues that are characteristic of GYF domains are indicated in red. (**C**) Morphology of four-week-old plants of Col-0, SALK_058114, *snc1* and SALK_058114 *snc1* double mutant. Bar = 1 cm. (**D**) Morphology of four-week-old plants of Col-0, SALK_035304, *snc1* and SALK_0353045 *snc1* double mutant. Bar = 1 cm.

SENSITIVE2 (FLS2) and Elongation Factor-TU RECEPTOR (EFR) recognize flagellin and EF-Tu respectively from bacteria and their functions require another RLK co-receptor, BRASSINOSTEROID INSENSITIVE1 – ASSOCIATED RECEPTOR KINASE1 (BAK1) (*Chinchilla et al., 2007*; *Monaghan and Zipfel, 2012*). From western blot analyses, we observed a slight increase in FLS2, but not significant difference in EFR and BAK1 protein levels in the *exa1* mutants (*Figure 9—figure supplement 2A–C*). MOS4 and MOS12 are involved in pre-mRNA splicing and proper splicing of *NLR* genes (*Xu et al., 2012*). The protein levels of MOS4 and MOS12 are comparable to that in the WT background (*Figure 9—figure supplement 2D and E*). The 70 kDa heat shock proteins (HSP70s) function as molecular chaperones to protect proteins against aggregation and are essential during normal growth (*Lin et al., 2001*). Histone proteins (H2A, H2B, H3 and H4) wrap DNA and form nucleosomes (*Peterson and Laniel, 2004*). Both HSP70s and Histone proteins are highly conserved and essential for plant growth. Interestingly, the H3, but not HSP70s, shows significant accumulation in *exa1* mutants (*Figure 9—figure supplement 2F and G*). Taken together, EXA1 does not seem to influence general protein synthesis. Besides NLRs, it also negatively impacts accumulation of other proteins such as FLS2 and H3. The specificity of EXA1 will be an interesting question to address in the future.

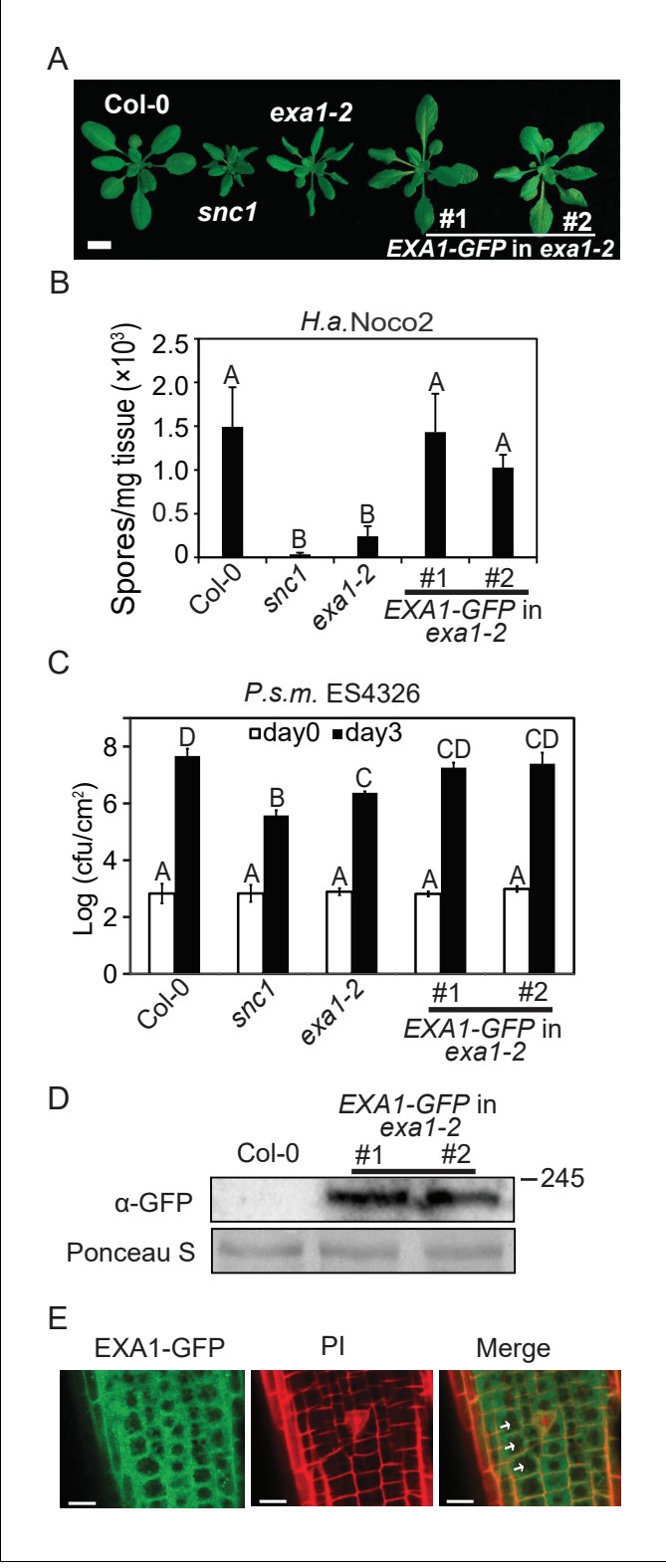

**Figure 6.** Subcellular localization of EXA1-GFP. (**A**) Morphology of four-week-old plants of Col-0, *snc1*, *exa1-2* and two independent lines of *EXA1-GFP* transformed into *exa1-2*. Bar = 1 cm. (**B**) Quantification of *H.a.* Noco2 sporulation on the indicated genotypes. One-way ANOVA was used to calculate the statistical significance among genotypes, which is indicated by different letters (p<0.01). Three independent experiments were carried out with similar results. Bars represent means ± SD (n = 4). (**C**) Growth of *P.s.m.* ES4326 on four-week-old leaves of the

*Figure 6 continued on next page*

*Figure 6 continued*

indicated genotypes at 0 and 3 dpi, with bacterial inoculum of $OD_{600}$ = 0.001. One-way ANOVA was used to calculate the statistical significance among genotypes, which is indicated by different letters (p<0.01). Three independent experiments were carried out with similar results. Bars represent means ± SD (n = 4). (**D**) Immunoblot analysis of EXA1-GFP expression in two independent transgenic lines. Equal loading is shown by Ponceau S staining of a non-specific band. Molecular mass markers in kilodaltons are indicated on the right. (**E**) Confocal images of EXA1-GFP fluorescence in root cells of an *exa1-2* transgenic line expressing *EXA1-GFP* under the control of its native promoter. Cell walls were stained using propidium iodide (PI). Arrows indicate cell nucleus. Merge means merged image between GFP and PI. Bar = 0.1 µm.

## EXA1 associates with the ribosome complex and eIF4E translation initiation factors

Regulators of translation often complex with ribosomes (*Ingolia, 2014*). To determine whether EXA1 acts at the translational level, we tested whether EXA1 associates with the ribosome through examining the interaction with RPL18, a ribosomal protein (*Zanetti et al., 2005*). A split-luciferase complementation assay was carried out where EXA1 and RPL18 were fused with the carboxyl-terminal (C-Luc) and amino-terminal (N-Luc) halves of the firefly luciferase, respectively (*Chinchilla et al., 2007*). As shown in *Figure 10A and B*, EXA1 indeed interacts strongly with RPL18. We further performed immunoprecipiration assays in *N. benthamiana* leaves which transiently expressed EXA1-FLAG and RPL18-HA, and in Arabidopsis plants stably transformed with both *EXA1-FLAG* and *RPL18-HA*. In both cases, RPL18-HA efficiently pulled down EXA1-FLAG (*Figure 10C and D*).

As EXA1 was predicted to contain a conserved motif for interaction with eIF4E, we also tested for interactions between EXA1 and eIF4E1 or eIF4E1B. Stronger luciferase activity was observed when C-Luc-EXA1 was co-infiltrated with eIF4E1-NLuc or eIF4E1B (*Figure 10E and F*), confirming the predicted interaction with the initiation factors. Interestingly, the interaction between C-Luc-EXA1 and mature SNC1-N-Luc was not detectable (*Figure 10G*), suggesting that EXA1 does not function at the termination step of translation. Together, these data confirm the sequence prediction that EXA1 is indeed associating with ribosomes and translational initiation factors, likely through translational repression of NLRs, as knocking out EXA1 leads to NLR accumulation.

## Discussion

To avoid autoimmunity, plant NLRs are under tight negative control through: (1) regulating appropriate *NLR* gene expression at the chromatin level (*Johnson et al., 2012*, *2015*), (2) proper alternative splicing of *NLR* mRNAs at the posttranscriptional level (*Johnson et al., 2016*), (3) coordination of N-terminal acetylation during NLR protein synthesis (*Xu et al., 2015*), and (4) post-translationally maintaining NLR protein homeostasis via the ubiquitin-mediated protein degradation pathway (*Cheng et al., 2011*; *Huang et al., 2014a*, *2014b*, *2016*). In this study, we identified a GYF domain protein, EXA1, which is involved in the negative regulation of NLR accumulation. Loss of EXA1 leads to enhanced disease resistance against both oomycete *H.a.* Noco2 and bacterial pathogen *P.s.m* ES4326, as well as increased NLR protein levels. Furthermore, EXA1 associates with the ribosome complex and likely exerts a role in translational repression.

GYF domain proteins are conserved in eukaryotes, serving as molecular adaptors (*Kofler and Freund, 2006*). The GYF sequence folds into a bulge-helix-bulge motif that binds to proline-rich repeats (*Freund et al., 2003*). Two types of GYF domains have been identified: CD2BP2-type and SMY2-type (*Kofler and Freund, 2006*), which mainly differ in that the SMY2-type contains an aspartate residue at position eight instead of a tryptophan in CD2BP2 (*Kofler and Freund, 2006*). Interestingly, although they share sequence similarities, these two types of GYF domain proteins seem to have distinct subcellular localizations and exert different functions (*Ash et al., 2010*). For example, several CD2BP2 type GYF domain proteins are found in the nucleus and participate in mRNA splicing while proteins of the SMY2-type GYF domains are mainly cytosolic and function in translational control.

CD2BP2 was the first GYF protein identified, interacting with CD2 and triggering T lymphocyte activation (*Nishizawa et al., 1998*). CD2BP2 binds to proline-rich regions of several proteins of the

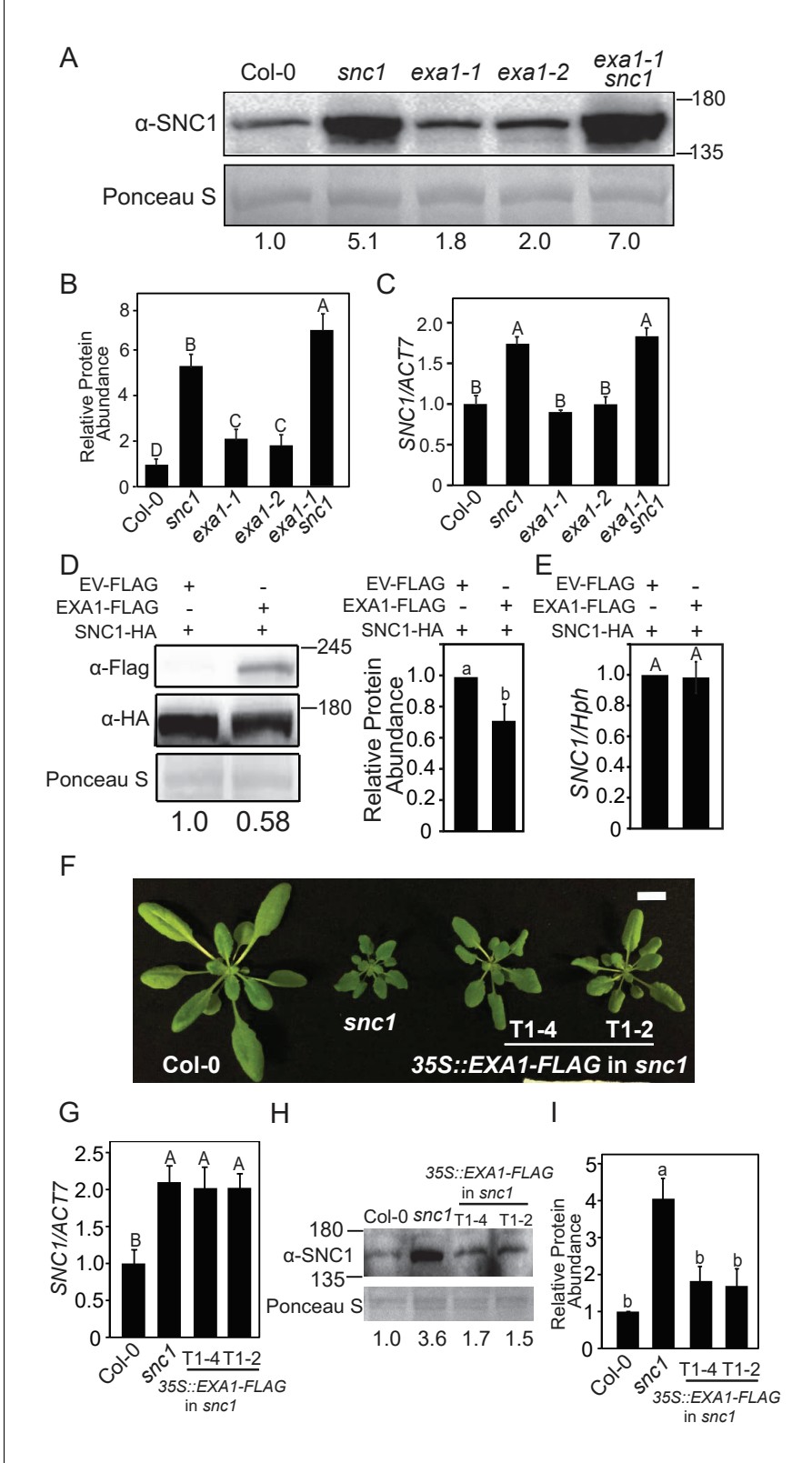

**Figure 7.** EXA1 negatively regulates SNC1 protein accumulation. (**A**) SNC1 protein levels in four-week-old soil-grown Col-0, *snc1*, *exa1-1*, *exa1-2* and *exa1-1 snc1* plants. Equal loading is shown by Ponceau S staining of a non-specific band. The relative SNC1 band intensity is shown below (normalized to loading control, relative to Col-0). Molecular mass markers in kilodaltons are indicated on the right. (**B**) Quantification of the relative abundance of SNC1 in **A**. Three independent experiments were carried out with similar results. Bars represent means ± SD (n = 3). Multiple pairwise t-test was used to

*Figure 7 continued on next page*

*Figure 7 continued*

calculate the statistical significance among genotypes, which is indicated by different letters (p<0.01). (C) *SNC1* transcript levels in four-week-old soil-grown plants of the indicated genotypes. *Actin7* was used to normalize the transcript levels. Three independent experiments were carried out with similar results. Bars represent means ± SD (n = 3). One-way ANOVA was used to calculate the statistical significance among genotypes, which is indicated by different letters (p<0.01). (D) SNC1-HA and EXA1-FLAG levels in *N. benthamiana* leaves expressing the indicated proteins. Equal loading is shown by Ponceau S staining of a non-specific protein band. The relative SNC1-HA band intensity is shown below (normalized to loading control, relative to control infiltration). Three biological repeats were carried out with similar results. Molecular mass markers in kilodaltons are indicated on the right. The quantification of all repeats is shown on the right as a bar graph. Bars represent means ± SD (n = 3). Multiple pairwise t-test was used to calculate the statistical significance among genotypes, which is indicated by different letters (p<0.05). (E) Transcript levels of *SNC1* in *N. benthamiana* leaves expressing the indicated proteins. Hygromycin resistance gene was used to normalize the transcript levels. Three biological repeats were carried out with similar results. Bars represent means ± SD (n = 3). One-way ANOVA was used to calculate the statistical significance among genotypes, which is indicated by different letters (p<0.01). (F) Morphology of four-week-old plants of Col-0, *snc1*, and two independent 35S::*EXA1-FLAG* transgenic lines in *snc1* background (T1-4 and T1-2). Bar = 1 cm. (G) *SNC1* transcript levels in four-week-old soil-grown plants of the indicated genotypes. *Actin7* was used to normalize the transcript levels. Two independent experiments were carried out with similar results. Bars represent means ± SD (n = 3). One-way ANOVA was used to calculate the statistical significance among genotypes, which is indicated by different letters (p<0.01). (H) SNC1 protein levels in four-week-old plants of the indicated genotypes. Equal loading is shown by Ponceau S staining of a non-specific band. The relative SNC1 band intensity is shown below (normalized to loading control, relative to Col-0). Molecular mass markers in kilodaltons are indicated on the left. (I) Quantification of the relative abundance of the SNC1 in H. Two independent experiments were carried out with similar results. Bars represent means ± SD (n = 2). Multiple pairwise t-test was used to calculate the statistical significance among genotypes, which is indicated by different letters (p<0.05).

splicing machinery to mediate protein-protein interactions in the spliceosome within the nucleus (*Kofler et al., 2004*). Knocking out CD2BP2 in mice greatly altered splicing, leading to growth retardation, defects in vascularization, and premature death. In yeast, one CD2BP2 type GYF protein, Lin1, was found to be a non-essential component of the U5 snRNP complex (*Stevens et al., 2001*). Lin1 also interacts with the cohesin complex component Irr1p and may play a role in mRNA splicing, DNA replication, and chromosome segregation (*Bialkowska and Kurlandzka, 2002*).

The SMY2 protein has been implicated in diverse biological pathways, such as vesicular transport (*Lillie and Brown, 1992*), mRNA surveillance (*Kofler et al., 2005*; *Georgiev et al., 2007*), membrane secretion (*Georgiev et al., 2008*; *Higashio et al., 2008*) and ribosome biogenesis (*Okano et al., 2015*). Mutations in another SMY2-type GYF protein GIGYF2 (Grb10-Interacting GYF Protein 2) have debatably been reported to associate with Parkinson's disease (*Lautier et al., 2008*; *Guo et al., 2009*; *Wang et al., 2010*, *2011*; *Zhang et al., 2015*). GIGYF2 interacts with the eukaryotic translation initiation factors and represses translation of a subset of mRNAs during embryonic development (*Morita et al., 2012*). In mice, disruption of such complex in the cytosol leads to increased translation and perinatal lethality.

EXA1 carries the SMY-2 type GYF domain and localizes to the cytosol (*Figures 4B* and *6E*). Our finding that EXA1 contributes to the negative regulation of NLR protein accumulations and associates with the ribosome complex suggests that EXA1 is a translational repressor, similar to GIGYF2. As enhanced resistance was observed in *exa1* mutants against diverse pathogens, including bacteria, oomycete, and viruses (*Hashimoto et al., 2016* and current study), it is possible that EXA1 represses multiple *NLR* genes and/or general defence positive regulators. Loss of EXA1 causes accumulation of these proteins, leading to enhanced general resistance. However, we cannot exclude the possibility that NLR proteins may also guard EXA1 and the loss of EXA1 triggers immune responses.

To our knowledge, this is the first report on GYF domain protein participating in regulating NLR levels. Our study has uncovered a previously unknown mechanism where NLR proteins are also negatively regulated at translational level and highlight the importance of such control for maintaining NLR homeostasis.

## Materials and methods

### Plant growth conditions and mutant screen

All *Arabidopsis thaliana* and *Nicotiana benthamiana* plants were grown under ambient conditions (16 hr light/8 hr dark, 22°C). The *muse* screen was described earlier (*Huang et al., 2013*).

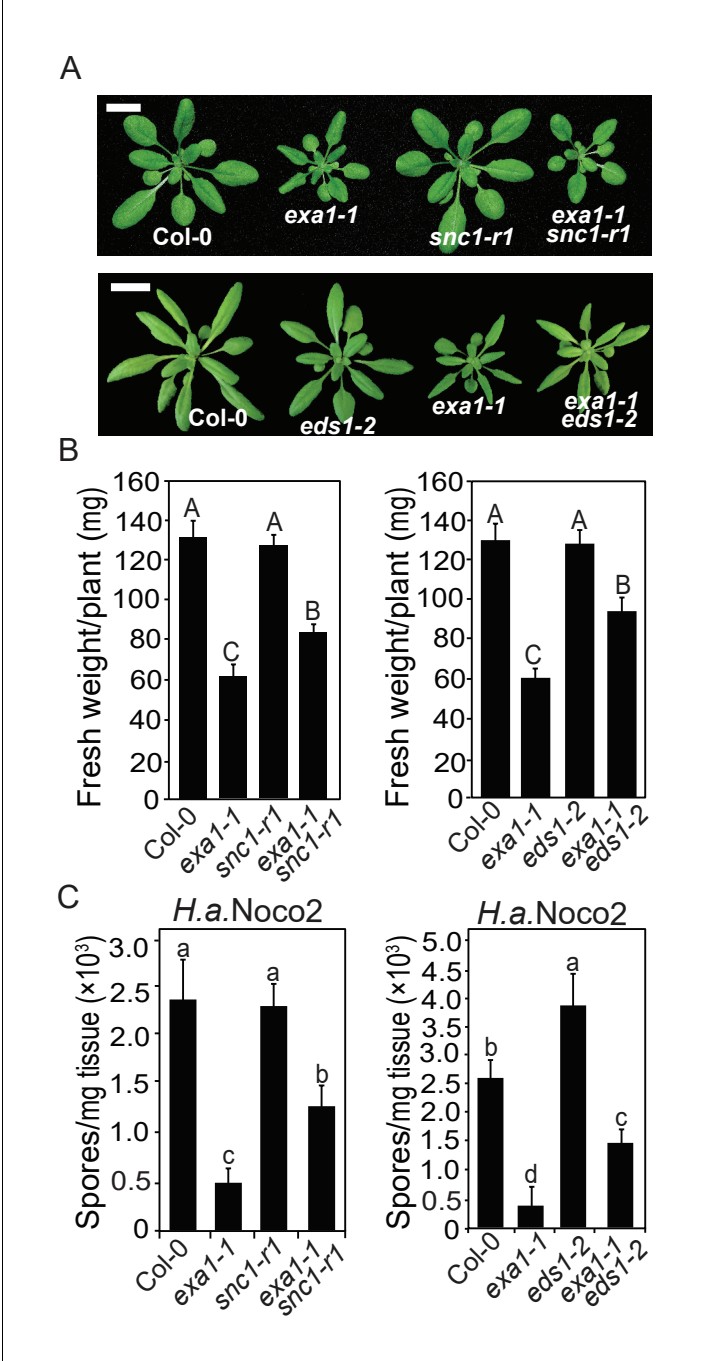

**Figure 8.** The autoimmunity of *exa1-1* partially depends on SNC1 and EDS1. (**A**) Morphology of four-week-old plants of the indicated genotypes. Bar = 1 cm. (**B**) Fresh weights of plants of A. One-way ANOVA was used to calculate the statistical significance, which is indicated by different letters (p<0.01). Bars represent means ± SD (n = 8). (**C**) Quantification of *H.a.* Noco2 sporulation on the indicated genotypes. One-way ANOVA was used to calculate the statistical significance, which is indicated by different letters (p<0.05). Two independent experiments were carried out with similar results. Bars represent means ± SD (n = 3).

## Positional cloning and next-generation sequencing

For positional cloning, homozygous *muse11-1 mos4 snc1* or *muse11-2 mos2 snc1 npr1* was crossed with L*er*. Thirty F2 plants from self-fertilized F1 plants exhibiting *snc1*-like or smaller-than-*snc1*

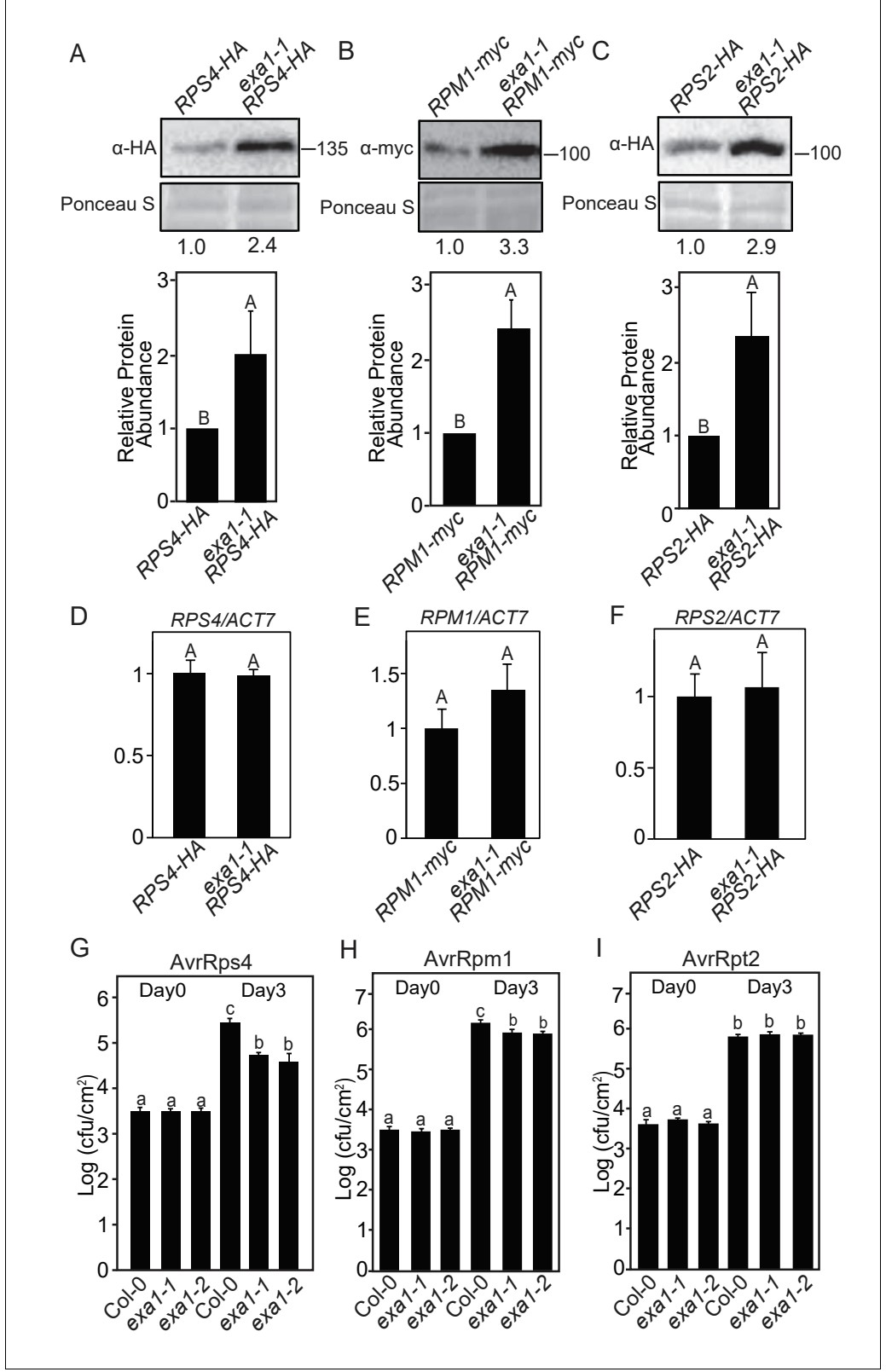

**Figure 9.** EXA1 affects the accumulation of NLR proteins RPS4, RPM1 and RPS2. (A–C) Immunoblot analysis of protein levels of RPS4-HA (A), RPM1-myc (B), RPS2-HA (C) in *exa1-1*. Equal loading is shown by Ponceau S staining of a non-specific band. Numbers underneath indicate the relative intensity of bands of RPS4-HA (A), RPM1-myc (B), and RPS2-HA(C) to a nonspecific band in Ponceau S staining. Molecular mass markers in kilodaltons are

*Figure 9 continued on next page*

*Figure 9 continued*

indicated on the right. The quantification of the bands is shown blow as a bar graph. Bars represent means ± SD (n = 3). One-way ANOVA was used to calculate the statistical significance among genotypes, which is indicated by different letters (p<0.01). (D–F) Transcript levels of *NLR* genes in four-week-old soil-grown plants of the indicated genotypes. *Actin7* was used to normalize the transcript levels. Two independent experiments were carried out with similar results. Bars represent means ± SD (n = 2). One-way ANOVA was used to calculate the statistical significance among genotypes, which is indicated by different letters (p<0.01). (G–I) Bacterial growth of avirulent pathogens *Pst* DC3000 AvrRps4 (G), *Pst* DC3000 AvrRpm1 (H) and *Pst* DC3000 AvrRpt2 (I) in Col-0 and two *exa1* alleles at 0 and 3 dpi with bacterial inoculum of $OD_{600}$ = 0.001. One-way ANOVA was used to calculate the statistical significance among genotypes, which is indicated by different letters (p<0.05). Three independent experiments were carried out with similar results. Bars represent means ± SD (n = 4).

The following figure supplements are available for figure 9:

**Figure supplement 1.** EXA1 does not affect the transcript levels of *NLR* genes *RPS4*, *RPM1* and *RPS2*.

**Figure supplement 2.** Protein accumulation of non-NLR proteins in *exa1* mutants.

morphology were used for crude mapping. For fine mapping, about five hundred F3 plants generated from F2 plants heterozygous for *muse11*, wild-type for *mos2* or *mos4*, and homozygous for *snc1* were used. After the *muse11* mutation was narrowed down between markers MUL8 (15.7 Mb) and K23L20 (18.02 Mb) on Chromosome 5, next-generation re-sequencing was performed to identify mutations within the *muse11* region. Plants homozygous for both *snc1* and *muse11* from seven individual mapping lines were pooled. About 10 g tissue was collected, and their nuclear genomic DNA was extracted and purified. The purified DNA was sequenced using Illumina whole-genome re-sequencing as previously described (*Huang et al., 2013*).The mutations in *AT5G42950* was confirmed by Sanger sequencing using 92–1 F and 92–1 R (*Supplementary file 1*).

## Expression analysis

About 0.05 g plant tissue was collected from four-week-old soil-grown plants and RNA was extracted using an RNA isolation kit (Bio Basic; Cat#BS82314). 0.4 µg of RNA was used to generate cDNA using ProtoScript II reverse transcriptase (NEB; Cat#B0368). Semi-quantitative PCR was performed as described before. Real-time PCR was performed using a Perfect Realtime Kit. Primers used for amplification of *PR1*, *PR2*, *ACTIN7*, *SNC1*, *RPS4*, *RPS2* and *RPM1* were described previously (*Zhang et al., 2003*; *Cheng et al., 2011*). Primers used for amplification of the transgenes are included *Supplementary file 1*.

## Pathogen infection assays

Two-week-old soil-grown seedlings were sprayed with *H.a.* Noco2 at a concentration of $10^5$ spores per ml of water. Infected plants were kept in a humid growth chamber (12 hr light/12 hr dark, 18°C). Sporulation was quantified at seven dpi using hemocytometers.

Four-week-old plants grown under normal growth conditions were used for bacterial infection assays. The procedure was described before (*Zhang et al., 2003*).

## Construction of plasmids and *Arabidopsis* transformation

Full-length *EXA1* genomic DNA including 1500 bp of native promoter sequence was amplified by primers 92–2 genomic-KpnI-F and 92–2 genomic-XbaI-R (*Supplementary file 1*) from WT genomic DNA. The amplified fragment was cloned into *pGreen0229-GFP* to generate *EXA1::EXA1-GFP* construct. This construct was electroporated into *Agrobacterium tumefaciens* GV3101 and transformed into *exa1-2* plants by floral dipping (*Clough and Bent, 1998*). The *EXA1* and *RPL18* coding sequences were amplified by primers MUSE11-KpnI-F, MUSE11-SmaI-R and RPL18- XbaI-F, RPL18-StuI-R to generate *pCambia1300-35S-MUSE11-FLAG* and *pCambia1300-35S-RPL18-HA* construct. These constructs were introduced into *A. tumefaciens*. Bacteria carrying these two constructs were co-transformed into *exa1-1* plants. Western blot analyses were used to identify T1 plants that contain both two transgenes.

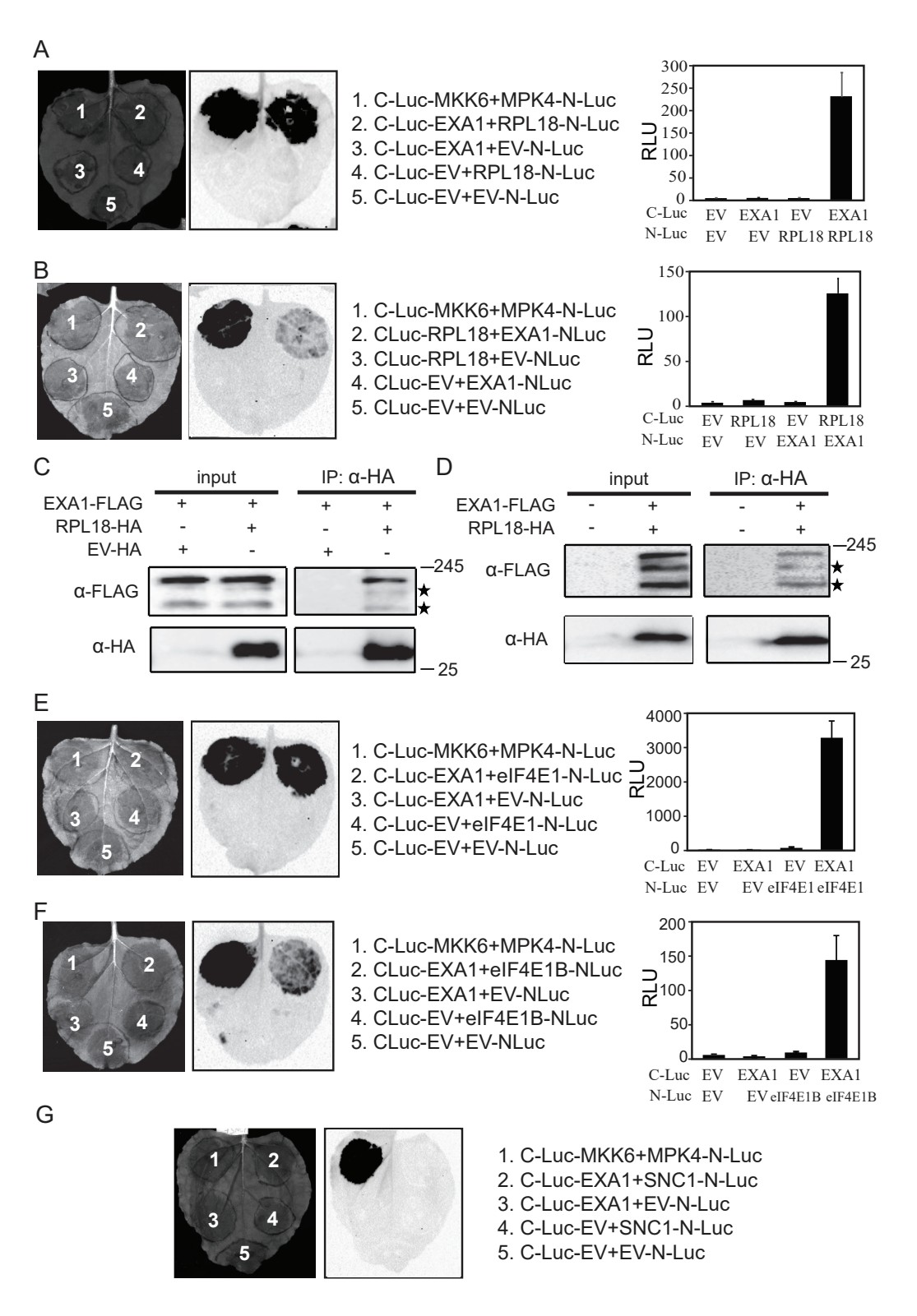

**Figure 10.** EXA1 interacts with ribosomal protein RPL18 and two eIF4E initiation factors eIF4E1 and eIF4E1B. (**A**) Interaction of EXA1 and RPL18 as tested by split-luciferase complementation assay in *N. benthamiana*. EXA1 was fused with C-Luc. RPL18 was fused with N-Luc. The empty N-Luc and C-Luc constructs were used as negative controls. The C-Luc-MKK6 and MPK4-N-Luc constructs were used as positive control. Three biological repeats were carried out with similar results. The relative luminescence units of all repeats are shown on the right as a bar graph. Bars represent means ± SD

*Figure 10 continued on next page*

Figure 10 continued

(n = 12). (B) Reciprocal split-luciferase complementation assay between EXA1 and RPL18. (C) Immunoprecipitation of EXA1-FLAG by RPL18-HA in *N. benthamiana*. Stars indicate EXA1-FLAG degration products. (D) Immunoprecipitation of EXA1-FLAG by RPL18-HA in Arabidopsis. Stars indicate EXA1-FLAG degration products. (E) Interaction of EXA1 and eIF4E as tested by split-luciferase complementation assay in *N. benthamiana*. Three biological repeats were carried out with similar results. The relative luminescence units of all repeats are shown on the right as a bar graph. Bars represent means ± SD (n = 12). (F) Interaction of EXA1 and eIF4E1B as tested by split-luciferase complementation assay in *N. benthamiana*. Three biological repeats were carried out with similar results. The relative luminescence units of all repeats are shown on the right as a bar graph. Bars represent means ± SD (n = 12). (G) Interaction of EXA1 and SNC1 as tested by split-luciferase complementation assay in *N. benthamiana*.

## Split luciferase complementation assay

The *EXA1* and *RPL18* coding sequences were amplified by primers MUSE11-KpnI-F, MUSE11-SalI-R and RPL18-KpnI-F, RPL18-SalI (no stop)-R to generate *pCambia1300-35S-Cluc- MUSE11* and *pCambia1300-35S-RPL18-NLuc* construct. These constructs were introduced into *A. tumefaciens*. Bacteria carrying these two constructs were co-infiltrated into four-week-old *N. benthamiana* leaves, and the infiltrated leaves were incubated with 1 mM luciferin two days later. Luminescence was recorded afterwards. Each bacterial strain was diluted to a final concentration of $OD_{600}$ = 0.4.

## Plant total protein extraction and immunoprecipitation

Plant total protein was extracted from about 100 mg tissue using a previously described protocol (*Huang et al., 2014a*). Extraction buffer was added to liquid nitrogen-ground powder and mixed well. The supernatant collected after centrifugation was transferred to a new tube, mixed with SDS loading buffer, and boiled at 95°C for 5 min. Proteins were separated on SDS-PAGE gel and immunoblotted by specific antibodies.

About 2 g *N. benthamiana* leaves expressing RPL18-HA with either EXA1-FLAG or EV-FLAG proteins or two-week-old plates-grown *Arabidopsis* seedlings expressing *RPL18-HA* and *EXA1-FLAG* was ground into powder with liquid nitrogen using a pre-chilled mortar and pestle. The HA-tagged proteins were immunoprecipitated using 20 µl HA beads (Roche) with gentle rotation for 2 hr at 4°C.

## Acknowledgements

Drs. Jane Parker, Brian Staskawicz, Jeff Dangl and Jian-Min Zhou are cordially thanked for sharing tagged NLR transgenic lines and *Pseudomonas* Avr strains. We thank Charles Copeland for critical reading of the manuscript. We would also like to acknowledge Ling Li and Yuelin Zhang for NGS sequencing. We appreciate Drs. Jianmin Zhou and Yuelin Zhang for providing antibodies against FLS2, EFR, BAK1, and MOS4. This work was financially supported by the Natural Sciences and Engineering Research Council of Canada (NSERC) Discovery Program and the Dewar Cooper Memorial Fund from UBC. ZSW, SH and DW were partly funded from the China Scholarship Council (CSC).

## Additional information

### Funding

| Funder | Grant reference number | Author |
| --- | --- | --- |
| Natural Sciences and Engineering Research Council of Canada | | Xin Li |
| University of British Columbia | Dewar Funds | Xin Li |

The funders had no role in study design, data collection and interpretation, or the decision to submit the work for publication.

### Author contributions

ZW, Data curation, Validation, Investigation, Methodology, Writing—original draft, Project administration; SH, Conceptualization, Data curation, Formal analysis, Investigation, Writing—original draft, Project administration; XZ, Formal analysis, Validation, Investigation, Methodology; DW, Validation,

Visualization, Methodology; SX, Conceptualization, Data curation, Formal analysis, Validation, Investigation, Methodology, Writing—original draft, Project administration; XL, Conceptualization, Data curation, Formal analysis, Supervision, Funding acquisition, Writing—original draft, Project administration, Writing—review and editing

## Author ORCIDs

Xin Li, iD http://orcid.org/0000-0002-6354-2021

## Additional files

### Supplementary files

• Supplementary file 1. Summary of primers used in this study

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
