## [Decision Letter]

Thank you for submitting your article "Regulation of plant immune receptor accumulation through translational repression by a GYF domain protein" for consideration by *eLife*. Your article has been favorably evaluated by Detlef Weigel (Senior Editor) and three reviewers, one of whom is a member of our Board of Reviewing Editors. The reviewers have opted to remain anonymous.

The reviewers have discussed the reviews with one another and the Reviewing Editor has drafted this decision to help you prepare a revised submission. We hope you will be able to submit the revised version within two months.

Summary:

The referees agree that your report on mechanism for negative regulation of NLR-type immune receptors and NLR-mediated immunity is intriguing and important. The identification of EXA1, which appears to be a part of the translational machinery, suggests that translational control contributes to immunity and regulates NLR-type immune receptor levels. Thus, your findings lend further support to the importance of negative regulatory principles in the control of immune activation in plants.

Essential revisions:

The referees, however, also state

1) That while there is a strong genetic correlation between the *exa1* mutant phenotype and enhanced NLR protein levels, further biochemical evidence would greatly strengthen your case (beside mere protein interaction studies in a heterologous system). In particular, it would be desirable to see complementary evidence for EXA1 interaction with ribosomal protein(s), preferably in *Arabidopsis*.

2) While it was shown that SNC1 protein but not its transcript is accumulating in the *exa1* mutant, the studies of the other NLRs were not accompanied by quantitative transcript analysis. This must be provided.

3) It remains unclear how specific the effects of EXA1on NLR stability are. Here, it would be important to know whether stability of PTI immune receptors or even non-immune proteins is affected by the exa1 mutation as well. In other words, is EXA1 a common regulator of protein stability or is its function restricted to NLRs? A broader role would not necessarily make EXA1 less interesting, but it would put the current results in a better perspective.

4) You should also elaborate on the possibility that NLR(s) might guard EXA1 and that impaired EXA1 (*exa1* mutation) triggers immune activation.

---

## [Author Response]

*Essential revisions:*

*1) That while there is a strong genetic correlation between the exa1 mutant phenotype and enhanced NLR protein levels, further biochemical evidence would greatly strengthen your case (beside mere protein interaction studies in a heterologous system). In particular, it would be desirable to see complementary evidence for EXA1 interaction with ribosomal protein(s), preferably in Arabidopsis.*

Immunoprecipiration assays were performed in *N. benthamiana* leaves which transiently expressed EXA1-FLAG and RPL18-HA, and in Arabidopsis plants stably transformed with both *EXA1-FLAG* and *RPL18-HA*. In both cases, RPL18-HA can efficiently pull down EXA1-FLAG (Figure 10).

*2) While it was shown that SNC1 protein but not its transcript is accumulating in the exa1 mutant, the studies of the other NLRs were not accompanied by quantitative transcript analysis. This must be provided.*

We tested the transcript levels of the other *NLR* genes in *exa1* mutant background (Figure 9 and Figure 9—figure supplement 1). No difference in transcript levels was observed compared with the wild-type background.

*3) It remains unclear how specific the effects of EXA1on NLR stability are. Here, it would be important to know whether stability of PTI immune receptors or even non-immune proteins is affected by the exa1 mutation as well. In other words, is EXA1 a common regulator of protein stability or is its function restricted to NLRs? A broader role would not necessarily make EXA1 less interesting, but it would put the current results in a better perspective.*

This is a very important point. We gathered many plant antibodies available to us and tested the protein levels of PTI immune receptors and non-immune proteins in the *exa1* mutants. We observed a slight increase in FLS2 and significant accumulation of H3 among seven proteins we tested. The other 5 proteins have comparable levels in *exa1* mutants as in WT (Figure 9—figure supplement 2). Taken together, EXA1 does not seem to influence general protein synthesis. However, besides NLRs, it also affects the accumulation of other proteins.

*4) You should also elaborate on the possibility that NLR(s) might guard EXA1 and that impaired EXA1 (exa1 mutation) triggers immune activation.*

We added in the Discussion that we cannot exclude the possibility that NLR proteins may guard EXA1 and the loss of EXA1 triggers immune responses..